# Activated Plasma Albumin Gel (APAG) in Transalveolar Technique for Maxillary Sinus Lift: A Case Series

Alessandro Leonida [1], Paolo Caccianiga [1,*], Ayt Alla Bader [1], Stefano Rosi [2], Saverio Ceraulo [1] and Gianluigi Caccianiga [1]

1  School of Medicine and Surgery, University of Milano-Bicocca, 20900 Monza, Italy
2  Independent Researcher, 60121 Ancona, Italy
*  Correspondence: p.caccianiga@campus.unimib.it

**Abstract:** Over the past 20 years, transalveolar techniques have progressively improved. They have become increasingly predictable and reliable, with the additional advantage of simplified procedures that are less operator dependent. The aim of this study is to evaluate the effectiveness of a new technique to lift the maxillary sinus through a transalveolar approach, Simple Minimal Safe (SMS), with use of activated plasma albumin gel (APAG). A total of 33 patients (22 female and 11 male), aged between 36 and 79, were consecutively operated on, with 44 implants positioned using the SMS technique. All were placed in the premolar or molar areas of the maxillary bone. No implant was lost during the follow-up period (6 months) and all implants were then prosthetically loaded. The average bone gain was 4.43 mm. In the first quadrant, sites 15, 16 and 17 were involved with an average bone gain of 3.5 mm, 4.6 mm and 4.5 mm, respectively. In the second quadrant the sites 24, 25, 26 and 27 were involved with an average bone gain of 4.25 mm, 4.5 mm, 4.4 mm and 4.5 mm, respectively. Analyzing the data considering the sex, implants in women had an average gain of 4.66 mm, while in men the average gain was 3.83 mm. With the SMS technique, we achieved a reduction in post-operatory morbidity and in the frequency of Schneiderian membrane perforation. In conclusion, maxillary sinus augmentation via the transalveolar approach has become a more predictable surgical procedure and an easier technique due to less operator-dependent processes.

**Keywords:** mini sinus-lift; maxillary sinus-lift; dental implant; APAG; concentrated grow factor; transalveolar technique; bone regeneration

## 1. Introduction

Following the loss of one or more dental elements in the lateral posterior regions of the upper jaw, the alveolar bone undergoes a vertical and horizontal contraction [1]. The resorption of the alveolar ridge may be furthermore associated to a parallel reduction of the residual bone caused by the expansion of the maxillary sinus due to the pneumatization phenomenon [2]. The limited availability of the residual bone and a poor bone quality often makes difficult the positioning of implants in these areas.

In case of atrophy of the upper jaw tied to the hyper-pneumatisation of the sinus with correct inter-arch ratios, the maxillary sinus floor lift represents the gold standard to restore the bone level in order to allow an implant-prosthetic rehabilitation with normal-sized implants.

In case, however, of vertical and/or transversal deficit, in association with extended pneumatisation of the sinus, it becomes necessary to restore the right inter-arch ratio. Therefore, regenerative bone techniques, such as guided bone regeneration (GBR), bone grafts, ridge expansion and, in more severe cases, revascularized grafts, are also needed [3].

The maxillary sinus lift was introduced by Boyne in the 1960s; however, only from the 1970s has it been proposed as a regenerative approach preliminary to implant-prosthetic rehabilitation in patients with largely pneumatized maxillary sinus and severe alveolar

reabsorption. The first scientific study on the maxillary sinus lift dates to 1980 [4], while the refinement of the technique is fulfilled many times latterly, thanks to Tatum [5].

The methodology foresaw, 3 months before the implant positioning, the access to the maxillary sinus through antrostomy in the antero-lateral wall of the maxillary, which allowed for detaching and for lifting a portion of the Schneider's membrane in order to later fill the subantral cavity with a graft of the only autologous bone taken from the iliac crest. Several recent studies confirm the reliability and long-term predictability of the maxillary sinus lift [6,7]. However, the invasiveness and the not so rare occurrence of intra- and post-operative complications related to the maxillary sinus augmentation via lateral approach have gradually led researchers and clinicians to develop alternative solutions, in particular in cases of monoedentulism with a residual bone height of 4–5 mm. In 1994, Summers proposed a one-stage technique to lift the maxillary sinus floor via the transalveolar approach, with two options: Osteotome Sinus Floor Elevation (OSFE) [8] and "Bone Added Osteotome Sinus Floor Elevation (BAOSFE) [9]. Both techniques relied on use of a sequence of osteotomes with concave tip that, once appropriately beaten with a surgical hammer, was able to produce a fracture in the apical direction of the cortical area of the maxillary sinus floor. The difference between the two methods is that the first aimed at exclusive use of the material obtained from the preparation of the osteotomic site, while the second also used grafting material that could be used repeatedly until the desired height was achieved. In 1995, Summers also proposed a two-stage technique (Future Site Development) [10] recommended in those cases where the residual ridge was not sufficient to guarantee the primary stability. This technique relied on the use of a coring mill to create a bone cylinder that was always lifted with osteotomes in order to lift the sinus membrane.

Thanks to the approach introduced by Summers, nowadays it is therefore possible to reduce the invasiveness of the intervention with several sinus lift techniques via the transalveolar approach. In fact, in the last 20 years, transalveolar techniques have considerably improved; furthermore, the procedures have been simplified in order to make them less operator dependent. These new methods have the purpose of preserving the integrity of the sinus membrane because it is considered the key factor for a successful lift and for a reduced occurrence of intra- and post-operative complications [11–13]. In fact, a non-perforated membrane is capable of guaranteeing protection and stability to the blood clot, which also results in a greater bone regeneration in the subantral region [14,15].

The aim of this study is to assess the effectiveness of a new technique for maxillary sinus lift via transalveolar approach, Simple Minimal Safe (SMS), using activated plasma albumin gel (APAG).

## 2. Materials and Methods

### 2.1. Sample Selection

A total of 33 patients (22 female and 11 male), aged between 26 and 79, were selected for the operation: 44 implants were positioned with the SMS technique, 32 on female patients and 12 on male patients (Table 1). All were placed in the premolar or molar areas of the maxillary bone (dental elements: 15, 16, 17, 24, 25, 26, 27). Inclusion and exclusion criteria were the following:

1.　Inclusion criteria:

- Sex: male or female;
- Age: between 26 and 78 years old;
- Systemic situation: patients should not have absolute contraindications to surgery; diabetics or pharmacologically compensated hyper-tense patients were included;
- Pharmacology: patients not under bisphosphonates or anticoagulant therapies;
- Smoking: non smokers;
- Stomagnathic situation: absence of periodontal illness without treatment or in active phase, excellent oral hygiene and general compliance, absence of acute injuries (abscess);
- Adequate thickness to perform the intervention with flapless technique.

2. Exclusion criteria:

- Age: below 20 and over 80 years old;
- Systemic situation: presence of pathologies that represent an absolute contraindication to surgery (IMA in the previous year, decompensated diabetes, hypertension without pharmacological treatment, chemotherapy or radiotherapy, in the district of our interest, in progress or during the preceding year, neoplasms);
- Pharmacology: use of bisphosphonates, anticoagulant or chemotherapy drugs;
- Smoking: The effect smoking may be deleterious to osseointegration and to the long-term survival of implants;
- Stomagnathic situation: periodontal illness without treatment or in active phase, poor oral hygiene and general compliance, presence of acute injuries (abscess);
- Adequate thickness to perform the intervention with flapless technique;
- Insufficient thickness for flapless intervention;
- Allergies: the patient must not be allergic to the molecules under consideration.

**Table 1.** Summary table showing the characteristics of the sample.

| | Sample | |
|---|---|---|
| Age Range | 36–79 years old | |
| Gender | Female | Male |
| Number of Patients | 22 | 11 |
| Number of Implants | 32 | 12 |
| Average Bone Gain | 4.66 mm | 3.83 mm |

*2.2. APAG Preparation*

After drawing two test tubes (white cap Vacuette) of blood from the patient, the test tubes were centrifuged in Medifuge 200 (Silfradent® srl, Forlì-Cesena, Italy). Once the centrifugation was complete, 0.5 cc of plasma was taken from both the test tubes at the white-red interface with two syringes. This allowed for the sucking of the rich portion of the hematopoietic stem cells CD34+ (15). In the following step, 4 cc of plasma was taken with two syringes identical to those used in the first step (Figure 1). The two latter syringes were placed inside the heater APAG (Silfradent® srl, Forlì-Cesena, Italy) and kept for 10 min at 75 °C. Thereafter, they were allowed to cool to ambient temperature for approximately 10–15 min. Using a specific valve, the agglutinated gel of the second syringe was mixed with the liquid containing CD34+ (Figure 2). This operation, repeated for approximately 2 min for 30 times, allowed for the APAG gel to load to the insulin syringe that was used for the Sinus Lift.

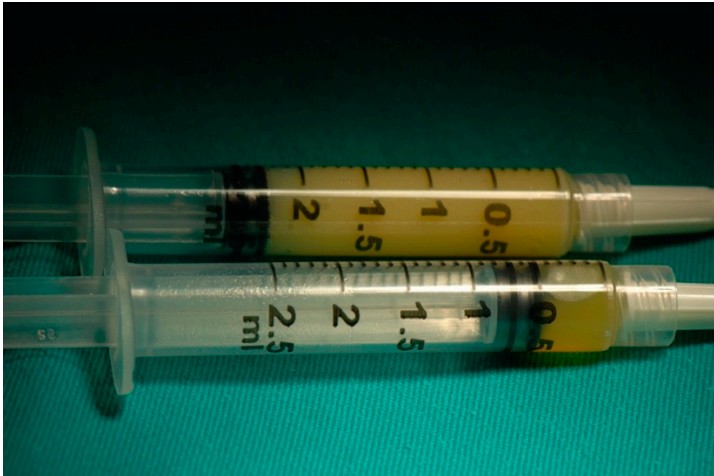

**Figure 1.** Two identical syringes: the first with a rich portion of the hematopoietic stem cells CD34+ and the second with 4 cc of plasma.

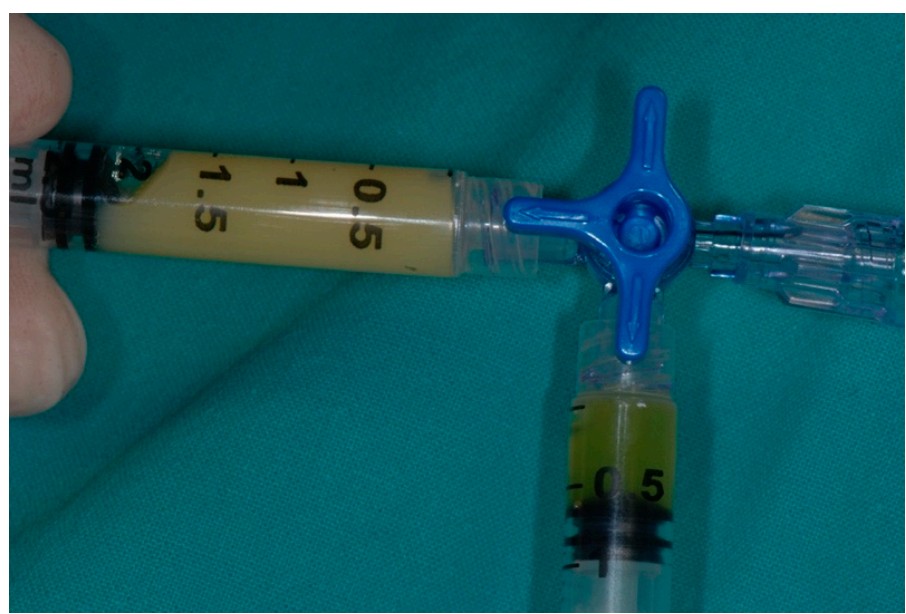

**Figure 2.** Using a specific valve, the agglutinated gel of the second syringe was mixed with the liquid containing CD34+.

### 2.3. Surgical Technique

The surgical operation was performed in flapless mode to facilitate the achievement of a seal at the time of the APAG injection, due to the presence of the mucosa.

Once the precise measurement of the distance between the alveolar ridge and the maxillary sinus floor with a CBCT (Pax 3D VATECH, Tecno-Gaz Spa®, Parma, Italy) (Figure 3) was taken, we used, at a very low rpm the first drill (2 mm in diameter) of the implanter Kit (Alpha-BioTec®, Petah Tikva, Israel) to perforate the sinus cortex. In all the interventions, we found evidence of perforation in the bleeding of the vascular septum below the Scnheider's membrane.

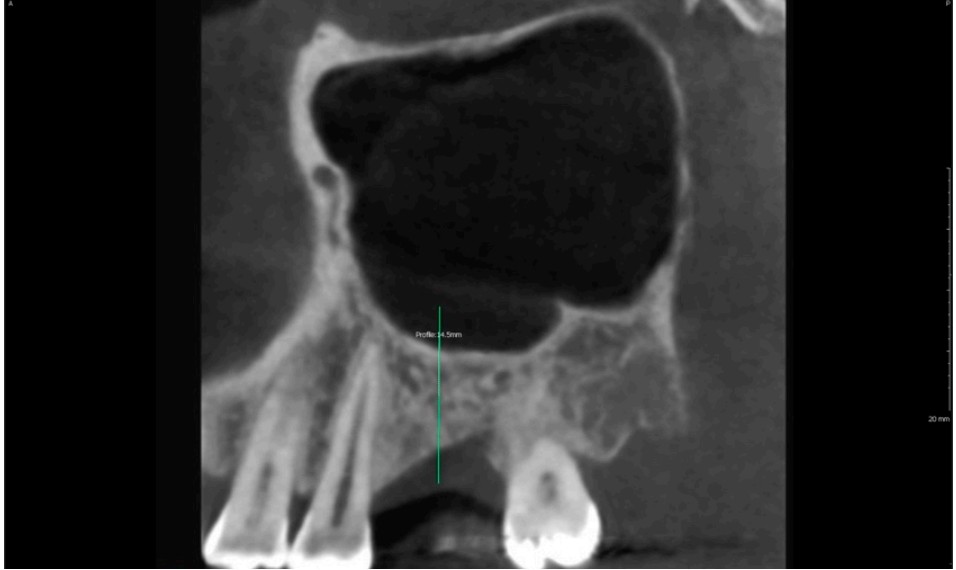

**Figure 3.** Sagittal section of CBCT-Taking the precise measurement of the distance between the alveolar ridge and the maxillary sinus floor with a CBCT.

We performed the Valsalva maneuver to seek confirmation of the integrity of the maxillary sinus membrane. Once we broke through the cortex at the foot of the sinus, we

injected the APAG gel (Figure 4) and then we positioned an SPI implant (Alpha-BioTec®, Petah Tikva, Israel) in complete safety.

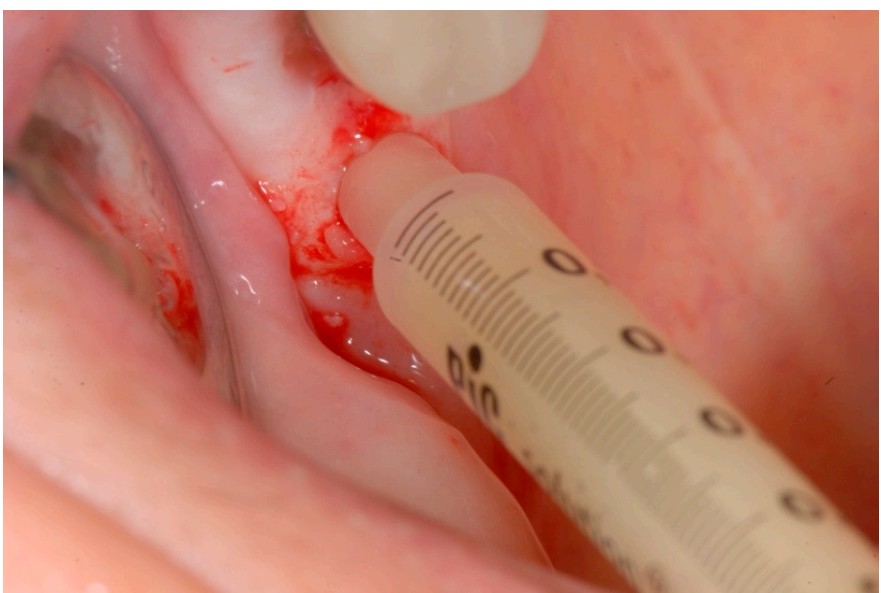

**Figure 4.** Injection of APAG gel through the prepared implant hole.

The transmucosal openings were closed either with a fragment of Buffy Coat CGF (Medifuge 200-Silfradent® srl, Forlì-Cesena, Italy) or re-positioning the bilaminar operculum taken with the mucotome.

### 2.4. Post-Operative Home Treatment

All treated patients were dismissed with the prescription of antibiotic therapy with Amoxicillin + ac. Clavulanic 1 g (one tablet every 12 h for 6 days). Once the surgery was completed, the patients were given an anti-inflammatory drug based on naproxen sodium 550 mg (Synflex) to avoid the onset of pain. They were advised to take other anti-inflammatories as needed. The recommendations prescribed to the patients were: do not blow the nose; do not use the mobile prosthesis immediately (it was repositioned three days later during the check-up).

### 2.5. Remote Follow-Ups

Follow-ups were carried out via CBCT (Pax 3D VATECH, Tecno-Gaz Spa®, Parma, Italy) 6 months later at "low resolution".

### 2.6. Case Study

RM: 68 years old, female, carrier of an upper skeletal prosthesis with attachments for lack of dental elements 14, 15, 16, 17 and 24, 25, 26, 27. The woman was eligible following the inclusion criteria shown above. We made a level one instrumental examination, orthomatomography, and then a CBCT (Pax 3D VATECH, Tecno-Gaz Spa®, Parma, Italy)) with which we have programmed the insertion of 6 SPI implants (Alpha-BioTec®, Petah Tikva, Israel) in the first and second quadrant in places 14, 16, 17 and 24, 26, 27. The area of the installations 16.17 and 26.27 had a deficit in height and for this reason an increase with SMS technology was necessary (Figure 5).

The surgery, after plexic anesthesia with mepivacaine hydrochloride 2% with adrenaline 1:100,000 (Septodont), was performed "flapless" with the SMS technique and the mucous membranes were filled with a fragment of Buffy Coat CGF (Medifuge 200-Silfradent® srl, Forlì-Cesena, Italy) (Figure 6). CBCT (Pax 3D VATECH, Tecno-Gaz Spa®, Parma, Italy)) was performed at low control resolution which highlighted the lifting of the Schnei-

der Membrane and the presence of a homogeneous and radio-opaque material below it (Figure 7a,b).

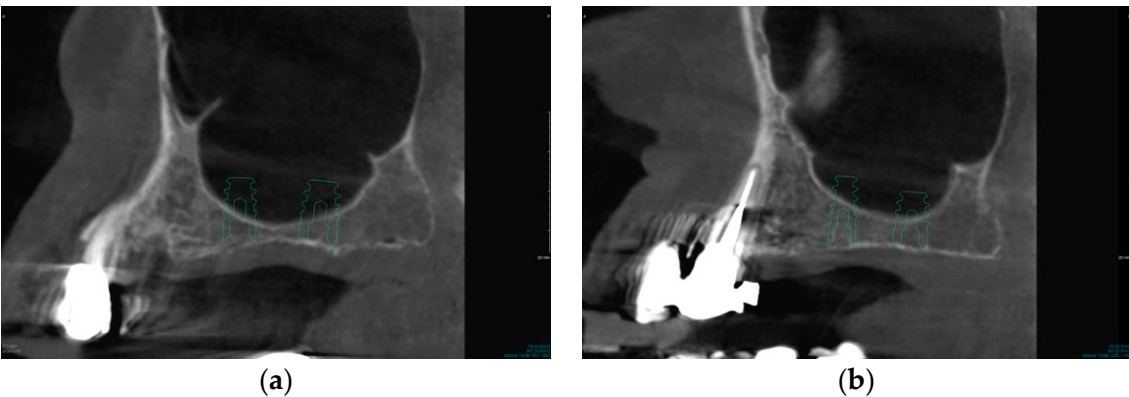

**Figure 5.** Sagittal section of CBCT—Pre-surgical case study with CBTC: (**a**) slide 1; (**b**) slide 2.

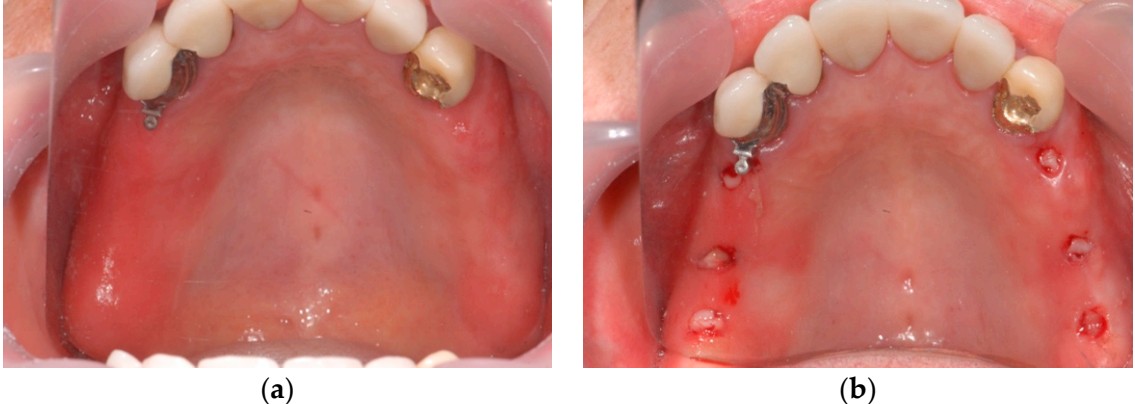

**Figure 6.** Clinical images: (**a**) pre-operative; (**b**) post-implant surgery.

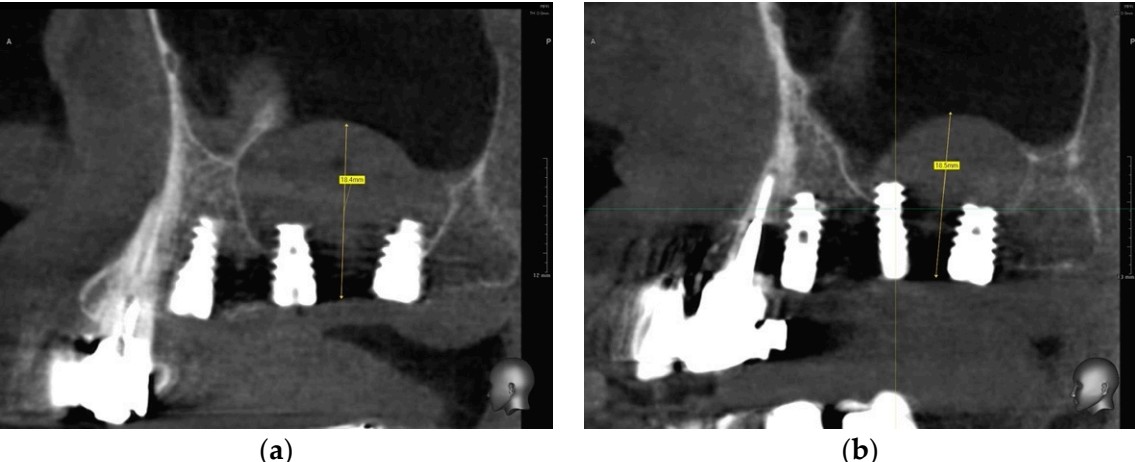

**Figure 7.** Sagittal section of CBCT—radiologic follow-up after implant surgery: (**a**) slide 1; (**b**) slide 2.

The patient was dismissed after prescription antibiotic therapy with Amoxicillin + ac. Clavulanic 1 gr (one tablet every 12 h for 6 days). Once the surgery was completed, the patient was given an anti-inflammatory drug based on naproxen sodium 550 mg (Synflex) to avoid the onset of pain. On the third day during the checkup, the wound showed no soft tissue tumefaction and no redness (Figure 8). The patient reported that the postoperative course was free of complications, without pain and swelling. This allowed her to carry out

her daily tasks normally, continuing her work even on the same day of the intervention. The patient always wore the mobile prosthesis properly unloaded in the following months.

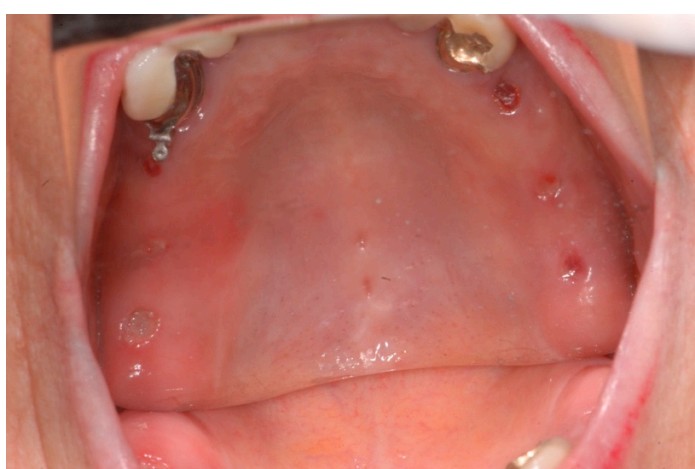

**Figure 8.** Clinical follow-up after three days.

At 6 months, we proceeded with the surgical re-entry to place the healing pillars and, after the soft tissue healing, the impression was taken with Trios intraoral optical scanner (3shape®, Milan, Italy). In a short time, using Dental System software (3shape®, Milan, Italy) with CAD-CAM technique, the prosthetic part consisting of customized abutments with Ti + Zirconium t-base and HIPC bridges (Bredent®, Bolzano, Italy) was designed and produced with a DWX-4 milling machine (Roland). (Figure 9a,b). At this point, a low resolution CBCT (Pax 3D VATECH, Tecno-Gaz Spa®,Parma, Italy)) control was performed (Figure 10a,b).

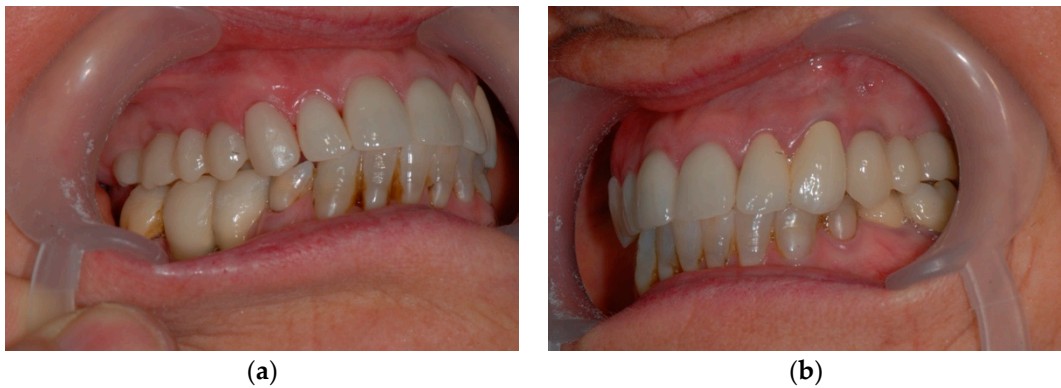

| (a) | (b) |

**Figure 9.** Prostheses defined in place: (**a**) I–IV quadrants; (**b**) II–III quadrants.

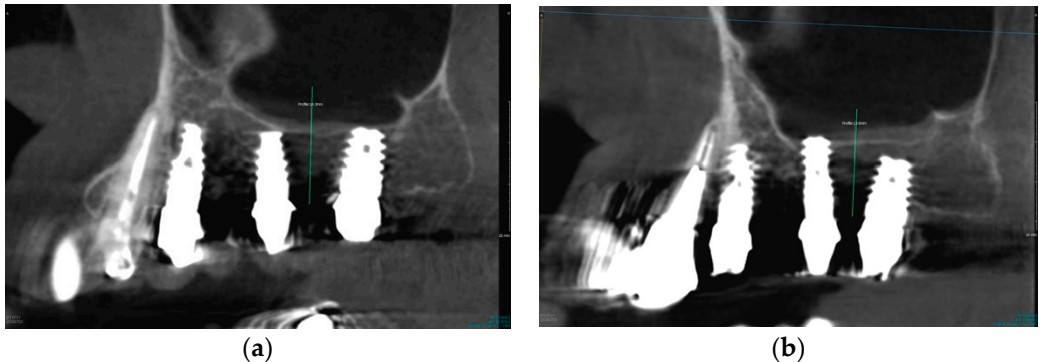

| (a) | (b) |

**Figure 10.** Sagittal section of CBCT—radiologic follow-up after 6 months: (**a**) slide 1; (**b**) slide 2.

### 3. Results

We placed 44 implants in 33 patients, using the SMS technique (Table 1). No implant was lost during the follow-up period (6 months) and all implants were prosthetically loaded. Prosthetic post-load follow-ups range from 6 to 12 months. There was no post-operative complication; no patient reported post-operative pain or swelling. There was no rupture of the sinus membrane. This was corroborated by Valsava's test that were consistently negative and post-operative radiographic checks.

The average bone gain was 4.43 mm. In the first quadrant, sites 15, 16 and 17 were involved with an average bone gain of 3.5 mm, 4.6 mm and 4.5 mm, respectively. In the second quadrant, sites 24, 25, 26 and 27 were involved with an average bone gain of 4.25 mm, 4.5 mm, 4.4 mm and 4.5 mm, respectively (Table 2). The results show how the majority of the sites have a gain between 4 and 5 mm. If we analyze the data by sex, we see that the 22 women received 32 implants with SMS technique and had an average gain of 4.66 mm, while the 11 men received 12 implants with an average gain of 3.83 mm. At level of each site, females in the first quadrant at position 16 and 17 had, respectively, an average bone gain of 4.95 mm and 4.5 mm. In the second quadrant at positions 24, 25, 26 and 27, there was an average bone gain of, respectively, 5 mm, 4.5 mm, 4.4 mm and 4.5 mm (Table 3). Males in the first quadrant, at positions 15, 16 and 17, had an average bone gain of 3.5 mm, 3.5 mm, 4.5 mm, respectively. In the second quadrant, at positions 24 and 26, they had an average bone gain of 2 mm and 4.37 mm (Table 4), respectively.

**Table 2.** Overall summary of patients.

| Patients | Gender | Age | Number of Implants | Site | Residual Bone Height | Implant | Increase Bone Post-Surgery | Bone Increase |
|---|---|---|---|---|---|---|---|---|
| 1 | f | 46 | 1 | 17 | 8 mm | 5.0 × 10 mm | 10 mm | 2 mm |
| 2 | f | 51 | 1 | 16 | 8 mm | 5.0 × 10 mm | 10 mm | 2 mm |
| 3 | f | 78 | 1 | 16 | 7 mm | 3.7 × 10 mm | 11 mm | 4 mm |
| 4 | f | 65 | 4 | 16 | 5 mm | 4.2 × 10 mm | 10 mm | 5 mm |
| | | | | 17 | 5 mm | 4.2 × 10 mm | 11 mm | 6 mm |
| | | | | 26 | 4 mm | 5.0 × 8 mm | 9 mm | 5 mm |
| | | | | 27 | 3 mm | 5.0 × 8 mm | 9 mm | 6 mm |
| 5 | m | 53 | 1 | 26 | 8 mm | 3.7 × 10 mm | 12 mm | 4 mm |
| 6 | f | 70 | 1 | 26 | 7 mm | 4.2 × 10 mm | 11 mm | 4 mm |
| 7 | f | 68 | 4 | 16 | 4 mm | 4.2 × 8 mm | 10 mm | 6 mm |
| | | | | 17 | 4 mm | 5.0 × 8 mm | 8 mm | 4 mm |
| | | | | 26 | 6 mm | 3.7 × 10 mm | 10 mm | 4 mm |
| | | | | 27 | 5 mm | 5.0 × 8 mm | 8 mm | 3 mm |
| 8 | f | 39 | 1 | 16 | 4.5 mm | 4.2 × 10 mm | 11 mm | 6.5 mm |
| 9 | f | 54 | 1 | 26 | 6.5 mm | 4.2 × 10 mm | 10 mm | 3.5 mm |
| 10 | f | 36 | 1 | 16 | 6.2 mm | 4.2 × 10 mm | 11 mm | 4.8 mm |
| 11 | m | 30 | 1 | 24 | 10 mm | 3.7 × 10 mm | 12 mm | 2 mm |
| 12 | f | 51 | 1 | 25 | 8 mm | 3.7 × 11.5 mm | 12 mm | 4 mm |
| 13 | f | 48 | 2 | 24 | 5 mm | 3.7 × 8 mm | 10 mm | 5 mm |
| | | | | 25 | 4 mm | 4.2 × 8 mm | 9 mm | 5 mm |
| 14 | m | 70 | 2 | 16 | 8 mm | 3.7 × 10 mm | 10 mm | 2 mm |
| | | | | 26 | 7 mm | 3.7 × 10 mm | 11 mm | 4 mm |
| 15 | m | 51 | 1 | 17 | 7 mm | 5.0 × 10 mm | 11 mm | 4 mm |
| 16 | f | 76 | 1 | 26 | 8 mm | 3.7 × 10 mm | 11 mm | 3 mm |
| 17 | m | 53 | 1 | 16 | 8 mm | 4.2 × 11.5 mm | 12 mm | 4 mm |
| 18 | m | 70 | 1 | 15 | 8.6 mm | 3.7 × 10 mm | 12 mm | 3.4 mm |
| 19 | m | 66 | 1 | 17 | 3 mm | 4.8 × 8 mm | 8 mm | 5 mm |
| 20 | f | 61 | 1 | 17 | 4 mm | 4.1 × 8 mm | 9 mm | 5 mm |
| 21 | f | 62 | 2 | 24 | 4 mm | 3.7 × 10 mm | 11 mm | 7 mm |
| | | | | 26 | 5 mm | 4.1 × 10 mm | 10 mm | 5 mm |
| 22 | f | 49 | 1 | 25 | 5.6 mm | 3.7 × 10 mm | 10 mm | 4.4 mm |
| 23 | f | 62 | 2 | 16 | 4 mm | 3.7 × 10 mm | 10 mm | 6 mm |
| | | | | 17 | 5.5 mm | 4.2 × 10 mm | 11 mm | 5.5 mm |

**Table 2.** *Cont.*

| Patients | Gender | Age | Number of Implants | Site | Residual Bone Height | Implant | Increase Bone Post-Surgery | Bone Increase |
|---|---|---|---|---|---|---|---|---|
| 24 | f | 72 | 1 | 16 | 4 mm | 5.0 × 8 mm | 8 mm | 4 mm |
| 25 | f | 57 | 1 | 26 | 7 mm | 5.0 × 10 mm | 11 mm | 4 mm |
| 26 | m | 79 | 1 | 16 | 8 mm | 5.0 × 10 mm | 11 mm | 3 mm |
| 27 | m | 56 | 1 | 26 | 7 mm | 5.0 × 10 mm | 10 mm | 3 mm |
| 28 | m | 62 | 1 | 16 | 5 mm | 5.0 × 10 mm | 10 mm | 5 mm |
| 29 | f | 79 | 1 | 16 | 6 mm | 3.7 × 8 mm | 8 mm | 2 mm |
| 30 | f | 71 | 1 | 26 | 6.3 mm | 4.2 × 10 mm | 11 mm | 4.7 mm |
| 31 | f | 54 | 2 | 24 | 8 mm | 3.7 × 10 mm | 11 mm | 3 mm |
|  |  |  |  | 26 | 4 mm | 4.1 × 10 mm | 10 mm | 6 mm |
| 32 | m | 40 | 1 | 26 | 6.5 mm | 5.0 × 10 mm | 11 mm | 6.5 mm |
| 33 | f | 66 | 1 | 16 | 1.2 mm | 5.0 × 10 mm | 10 mm | 8.8 mm |
|  |  |  | 44 Implants |  |  |  |  |  |

**Table 3.** Summary table of the female sample.

| Sites | 17 | 16 | 15 | 24 | 25 | 26 | 27 | |
|---|---|---|---|---|---|---|---|---|
| Number of Implants | 5 | 10 | 0 | 3 | 3 | 9 | 2 | 32 Total of implants |
| Bone Gain | 4.5 mm | 4.95 mm | 0 | 5 mm | 4.5 mm | 4.38 mm | 4.5 mm | 4.66 mm Average bone gain |

**Table 4.** Summary table of the male sample.

| Sites | 17 | 16 | 15 | 24 | 25 | 26 | 27 | |
|---|---|---|---|---|---|---|---|---|
| Number of Implants | 2 | 4 | 1 | 1 | 1 | 0 | 4 | 12 Total of implants |
| Bone Gain | 4.5 mm | 3.5 mm | 3.5 | 2 mm | 4.5 mm | 0 | 4.37 mm | 3.83 mm Average bone gain |

The characteristics of the implants and the levels of bone gain obtained with this technique, differentiated by dental element and sex of the patient, are highlighted in Tables 1–4.

## 4. Discussion

The effectiveness of sinus lift for the rehabilitation of atrophic posterior jaw has been widely documented in the literature [16]. However, the high post-operative morbidity and the risk of developing post-operative complications have prompted the research to develop progressively less invasive methods to elevate the floor of the maxillary sinus: first introducing the transalveolar technique and then developing methods that increasingly aim at atraumaticity. In transalveolar sinus lift, similar to the traditional lift with lateral anthrostomy, the most frequently reported complication in the literature is the perforation of the Schneiderian membrane [17]. With the new minimally invasive techniques of transalveolar lift, there has been a decrease in the frequency of drilling, which settles around an average value of 3.8% for the transalveolar lift [17], while in the lateral lift it is approximately five times more frequent. Membrane integrity is a key factor for the survival of the bone graft and implant. The perforation of the Schneider membrane is linked to a higher incidence of post-operative complications, such as graft failure and infection [11–13]. In addition, the size of the perforation is inversely proportional to implant survival [12].

However, there is no strong evidence that intraoperative perforations in the transalveolar lift lead to an increased risk of post-operative complications. This is due to the fact that

only small ruptures of the membrane can be managed by transalveolar access; the eventual perforation generally determines a change in the therapeutic plan with the insertion of an implant of reduced length where possible or with a surgery with lateral access to the maxillary sinus that allows for managing the rupture of the membrane.

Although Garbacea et al. [18] state that the incidence of perforations in the transalveolar rise is often underestimated because the Valsalva's test alone is not considered completely reliable to verify the integrity of the membrane, the low incidence of post-operative complications suggests that the maneuver is reliable to detect at least medium-large perforations. The perforations of small size and those properly managed, avoiding the use of biomaterial that can be displaced in the sinus through the laceration, seem therefore not to affect the post-operative course.

The addition of a biomaterial in the subantral cavity in the transalveolar lift is still under discussion today. As early as 1993 Boyne [19] had shown how it was possible to add new bone around the apex of implants placed in the elevated maxillary sinus without the addition of biomaterial.

However, this was only possible when the implant was only 2 to 3 mm above the floor of the maxillary sinus. Instead, when the implant extended 5 mm beyond the original sinus level, only in 50% of cases was it possible to appreciate the formation of new bone around the apex of the implant. According to Palma [20] and Leonida [21], in fact, the Schneiderian membrane would also have osteoinductive properties.

Several authors do not use any biomaterial or even a cushion agent, i.e., a material with the function of maintaining space and preventing membrane collapse, and they obtain satisfactory results [20,21]. Pjetursson et al. [7] compared bone remodeling of implants inserted following transalveolar lift without the addition of any material or with the addition of deproteinization bovine bone.

In 2017, Gastaldi et al. evaluated the efficacy of short (5 or 6 mm long) dental implants vs. 10 mm or longer implants placed in crestally lifted sinuses. Both techniques achieved excellent results and no differences were observed between prostheses supported by one to two 5 or 6 mm long implants vs. 10 mm long in posterior atrophic maxillae up to 3 years after loading; therefore, it is up to clinicians to decide which procedure to use [22].

The average height gain of alveolar bone was 4.1 mm in cases with biomaterial, which is significantly higher than 1.7 mm in cases of lift without grafting. When a transalveolar sinus lift is performed without biomaterial insertion, only modest new bone formation can be expected compared to when a bone graft is inserted. Some authors prefer not to place bone grafting in the subantral cavity, entrusting the scaffold and space making effect to a sponge of radiopaque collagen to be visible on radiography [23]. However, other authors use platelet concentrates as a graft, preventing perforations due to the lack of rough surfaces in contact with the membrane [20,21]. In addition, the latter also have the advantage of detaching and lifting the membrane from the floor of the sinus in a less traumatic way by exploiting the hydraulic pressure.

Many authors agree that a residual bone height of at least 4–5 mm [13,24] should be considered as a clinical indication for the transalveolar lift of the floor.

In fact, when the ridge is severely reabsorbed, below 4 mm, a transalveolar sinus lift would elevate the membrane beyond its physiological limits, which according to Nkenke et al. [23] would be 3 mm, while other studies report 5 mm as the membrane elevation limit [18,25].

Pommer et al. [26] have shown in an in vitro study that the Schneiderian membrane can be stretched on average up to 132.6% of its original size before rupturing. Regardless of the technique used, the authors believe that during transalveolar sinus floor elevation, a proper circumferential disconnection of the membrane prior to its elevation is essential to reduce its adhesion to the bone floor and its resistance to elevation, thus decreasing the risk of perforation during lifting.

In 2014, Esposito et al. compared the effectiveness of two different techniques to lift the maxillary sinus via a crestal approach: the Summers technique versus the Cosci technique. All 15 partially edentulous patients were followed to 3 years after implant loading. There

were no statistically significant differences for marginal bone level changes between the two groups. Both crestal sinus lift techniques (Cosci technique and Summers technique) produced successful results over a 3-year follow-up period, but the Cosci technique required less surgical time, determined less intra- and postoperative morbidity and was preferred by patients [27].

The subject of future studies must be the maintenance of implants inserted thanks to bone regeneration techniques, such as that proposed in this study. A literature review by Butera and Scribante et al. [28] had the purpose of investigating if the use of lasers, ozone, probiotics, glycine and/or erythritol, and chlorhexidine in combination with non-surgical peri-implant treatment have additional beneficial effects on the clinical parameters [29,30]. Most of the studies did not show any additional benefit of these therapies in the evaluation of bleeding on probing, probing pocket depth, or plaque index, among the proposed treatments. For example, a study by Colombo and Scribante et al. [31] revealed that the use of ozonized gel in addition to the standard non-surgical therapy generally did not significantly differ if compared to the use of chlorhexidine. The use of laser was the most studied in the literature, with the achievement of a reduction of bleeding and pocket depth. There are several works in literature about the efficacy of lasers and LEDs, not only in oral surgery but even in other fields of dentistry, especially in the management of periodontitis and periimplantitis and in orthodontics, [32–41]. This success is explained by a series of characteristics specific to laser technology, in fact, thanks to the photoacoustic, photochemical, photothermal and photomechanical properties, the laser makes it possible to reduce bacterial load at the intervention site [32–37]. Another important feature is the photobiomodulation properties of laser and LEDs; at the bone level, it promotes the healing process in the post-surgical period and reduces post-surgical pain [38], as well as the orthodontic pain [39–42].

## 5. Conclusions

In line with modern dentistry, which is evolving in all its disciplines towards less invasive and more simplified approaches, the described method of transalveolar elevation Simple Minimal Safe (SMS) represents a step in this direction. In fact, if the integrity of the sinus membrane represents the foundation of mini-invasiveness in sinus lift, this technique fully respects it.

The study has shown a reduction in post-operative morbidity and the incidence of perforation of the Schneiderian membrane, as well as making transalveolar sinus lift an increasingly predictable surgery and within the reach of a greater number of clinicians due to simple and minimally operator-dependent processes. Although the average bone gain data are comparable and in some cases even greater than the techniques present in the literature, the authors recommend a series of randomized clinical trials for greater validation of the technique and its results.

**Author Contributions:** Conceptualization, A.L.; Data curation, P.C.; Formal analysis, A.L.; Investigation, S.R.; Methodology, G.C.; Project administration, G.C.; Resources, G.C.; Software, A.A.B.; Supervision, G.C; Validation, P.C.; Visualization, S.C.; Writing—original draft, A.L.; Writin—review & editing, P.C. and A.A.B. All authors have read and agreed to the published version of the manuscript.

**Funding:** This research received no external funding.

**Informed Consent Statement:** Informed consent was obtained from all subjects involved in the study.

**Data Availability Statement:** Not applicable.

**Conflicts of Interest:** The authors declare no conflict of interest.

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
