# Peer review of "Activated Plasma Albumin Gel (APAG) in Transalveolar Technique for Maxillary Sinus Lift: A Case Series"

_inventions, doi:10.3390/inventions7040099_

Round 1

Reviewer 1 Report

Dear Authors,

The article: 'Activated Plasma Albumin Gel (APAG) in Transalveolar Technique for Maxillary Sinus Lift: a Cases Series' was to evaluate the effectiveness of a new technique to lift the maxillary sinus through a transalveolar approach, Simple Minimal Safe (SMS), with use of Activated plasma albumin gel (APAG).

English language and style must be corrected.

Lot punctuation mistakes should be corrected. 

e.g. abstract: 4,43 should be 4.43 

In English, the hundredths and decimal parts are written after the point. Correct throughout the text, there are numerous spelling errors.

Are the authors of the article students? There is no affiliation department. Do students perform such procedures? Who performed the surgical procedures?

Introduction is crealy written

Materials and methods:

Sentences do not begin with numbers. This is an invalid notation (33 patients (22 female and 11 male), aged between 36 and 79, were operated. 44 implants were positioned with SMS technique (Tab 1).

Data mismatch - once the patients are aged 26-79, then 20-80 !!!!

Enter the names of countries and cities for the devices.

Figure 1-3 - correct - the numbers are written twice.

Add information about the REC.

For CBCT figures - add the name of the section

Correct tables 1-4 - notation of numbers. Prepare tables according to MDPI guidelines.

Add a table with abbreviations.

References should be prepared in accordance with the MDPI guidelines

To sum up, article should be reconsider after major revision.

Author Response

Dear reviewer,

In responses to your review, thank you for your work on our article 'Activated Plasma Albumin Gel (APAG) in Transalveolar Technique for Maxillary Sinus Lift: a Cases Series'.

We changed the file according to your indications and as follows:

  • We checked and corrected the English language, spelling errors and punctuation mistakes;
  • We corrected al the numbers with decimal parts. 4,43 into 4.43 in the abstract;
  • We changed the introduction as you suggested
  • For material and methods, we have:
    • adjusted phrases that started with the numbers was overhauled.
    • adjusted data mismatch
    • put the names of countries and cities for the devices.
    • corrected the numbers under the figures
    • added the type of section in the figure of the CBCT
    • aorrected the numbers in the tables 1-4
    • added a table with abbreviations.
    • checked the references with the MDPI guidelines

About your questions:

"Are the authors of the article students? There is no affiliation department. Do students perform such procedures? Who performed the surgical procedures?"

The department affiliation is: School of Medicine and Surgery, University of Milano-Bicocca, 20900 Monza, Italy

Authors: Prof. Dr. Gianluigi Caccianiga is researcher and adjunt professor at Milano-Bicocca University, Dr. Alessandro Leonida and Dr. Saverio Ceraulo are contract professors at Milano-Bicocca University, Dr. Paolo Caccianiga and Dr. Ayt Alla Bader are recently graduated and now visiting doctors at Milano-Bicocca University, Dr. Stefano Rosi is a private practicioner.

The surgical procedures were performed by the teachers with the assistance of former students.

About the informations you asked for about the REC:

This article presents a series of clinical cases, which we have been able to create thanks to the approval of the Istituto Superiore di Sanità for carrying out experimentations (Prot. 30/07 / 2007-0040488), and to the informed consent obtained from patients, does not require the approval of an ethics committee as it is not a clinical trial study.

Thank you for you time.

We hope that our paper will meets your expectations and will be published on Inventions journal.

Best regards,

Dr. Paolo Caccianiga

Dr. Ayt Alla Bader

Reviewer 2 Report

This is a case series on a crestal sinus lift approach using a plasma gel as biomaterial. The topic could be of interest, however I would arise the followings:

- Please check for typing mistakes (e.g. Pag. 2 "surhery")

- I would ask the Authors to better underline the possible limitations of this study such as the study design, smokers exclusion etc.

- I would avoid to use terms like "demonstrated" in the conclusions being just a case series. To demonstrate a hypothesis RCTs are needed. For this reason I would suggest the Authors to take into consideration the studies: PMID: 24977247 and PMID: 29234746 in the discussion.

Author Response

Dear reviewer,

In responses to your review, thank you for your work on our article 'Activated Plasma Albumin Gel (APAG) in Transalveolar Technique for Maxillary Sinus Lift: a Cases Series'.

We changed the file according to your indications and as follows:

  • We checked the typing mistakes
  • We underlined the possible limitations of this study such as smokers exclusion, like you said.
  • We removed "demonstrated " to use a more consonant synonym as “shown”
  • We take into consideration the studies: PMID: 24977247 and PMID: 29234746 in the discussion as you suggested.

We hope that our paper will meet your expectations and will be published on Inventions journal.

Thank you for you time.

Best regards,

Dr. Paolo Caccianiga

Dr. Ayt Alla Bader

Reviewer 3 Report

Manuscript of considerable interest for the dental sector, it needs a major revision before proceeding with its publication

Abstract, it highlights the data collected more clearly

Few keywords, insert specific ones

Introduction: well described

Materials and methods: there is a lack of sample numbers and home treatment

Results: very confusing, making them more usable to the reader, highlighting the statistically significant data

Discussion: to add as future objectives the implant maintenance using natural substances such as (probiotics, post biotics, para probiotics, ozonated water, ozone, laser) already studied by the research group of Prof. Scribante et al.

Author Response

Dear reviewer,

Thanks for your kind suggestions, we tried to apply them to improve this article:

  • Abstract: we have expanded it, presenting the data in a more complete way;
  • Keywords: we have added some;
  • Introduction: thank you;
  • Materials and methods: we have made the characteristics of the samples clearer and we have clarified the management of the post-operative and at home by the patient;
  • Results: we highlighted that the results were differentiated according to the sites of the teeth and the sex of the patients using the tables. We have not highlighted statistically significant data, because statistical analyzes have not been performed, since the article is a presentation of a case series;
  • Discussion: we have expanded it, adding the part you suggested, with reference to the works of Prof. Scribante and also to our works on the laser.

We hope that our work meets your expecations and can be considered to be published on Inventions.

Faithfully,

Dr. Paolo Caccianiga

Round 2

Reviewer 1 Report

Accept after minor revision (corrections to minor methodological errors and text editing)

Author Response

Dear reviewer,

Thank you for your kind review. We have made minor revisions.

Faithfully,

Dr. Paolo Caccianiga

Reviewer 3 Report

The manuscript has been correctly revised, it can be submitted for publication

Author Response

Dear reviewer,

Thank you for your kind review.

Faithfully,

Dr. Paolo Caccianiga